

# Transfer of diazotroph derived nitrogen towards non-diazotrophic planktonic communities: a comparative study between *Trichodesmium erythraeum*, *Crocosphaera watsonii* and *Cyanothece* sp.

H. Berthelot[1], S. Bonnet[1,2], O. Grosso[1], V. Cornet[1], and A. Barani[1]

[1]Aix Marseille Université, CNRS/INSU, Université de Toulon, IRD, Mediterranean Institute of Oceanography (MIO) UM 110, 13288, Marseille, France
[2]Institut de Recherche pour le Développement, CNRS/Aix-Marseille Université, Mediterranean Institute of Oceanography (MIO), 101 Promenade R. Laroque, BPA5, 98848, Noumea cedex, New Caledonia

Received: 27 November 2015 – Accepted: 29 November 2015 – Published: 15 January 2015

Correspondence to: H. Berthelot (hugo.berthelot@mio.osupytheas.fr)

Discussion Paper | Discussion Paper | Discussion Paper | Discussion Paper |

**BGD**

doi:10.5194/bg-2015-607

**Transfer of diazotroph derived nitrogen**

H. Berthelot et al.



## Abstract

Biological dinitrogen ($N_2$) fixation is the major source of new nitrogen (N) for the open ocean, and thus promotes marine productivity, in particular in the vast N-depleted regions of the surface ocean. Yet, the fate of the diazotroph-derived N (DDN) in marine ecosystems is poorly understood and its transfer to auto- and heterotrophic surrounding plankton communities is rarely measured due to technical limitations. Moreover, the different diazotrophs involved in $N_2$ fixation (*Trichodesmium* spp. vs. UCYN) exhibit distinct patterns of $N_2$ fixation and inhabit different ecological niches, thus having potentially different fates in the marine food webs, that remains to be explored. Here we used nanometer scale secondary ion mass spectrometry (nanoSIMS) coupled with $^{15}N_2$ isotopic labelling and flow cytometry cell sorting to examine the DDN transfer to specific groups of natural phytoplankton and bacteria during artificially-induced diazotroph blooms in New Caledonia (southwestern Pacific). The fate of the DDN was compared according to the three diazotrophs: the filamentous and colony forming *Trichodesmium erythraeum* (IMS101), and the unicellular strains *Crocosphaera watsonii* WH8501 and *Cyanothece* ATCC51142. After 48 h, 7–17 % of the $N_2$ fixed during the experiment was transferred to the dissolved pool and 6–12 % was transferred to non-diazotrophic plankton. The transfer was twice as high during the *T. erythraeum* bloom than during the *C. watsonii* and *Cyanothece* blooms, arguing that filamentous diazotrophs blooms are more efficient at promoting non-diazotrophic production in N depleted areas. The amount of DDN released in the dissolved pool did not appear as a good indicator of the DDN transfer efficiency towards the non-diazotrophic plankton. In contrast, the $^{15}N$-enrichment of the extracellular ammonium ($NH_4^+$) pool was a good indicator of the DDN transfer efficiency: it was significantly higher in the *T. erythraeum* than in unicellular diazotroph blooms, leading to a DDN transfer twice as efficient. This suggests that $NH_4^+$ was the main pathway of the DDN transfer from diazotrophs to non-diazotrophs. The three simulated diazotroph blooms led to significant increases in non-diazotrophic plankton biomass. This increase in biomass was first associated with heterotrophic

Discussion Paper | Discussion Paper | Discussion Paper | Discussion Paper |

## BGD

doi:10.5194/bg-2015-607

**Transfer of diazotroph derived nitrogen**

H. Berthelot et al.

bacteria followed phytoplankton, indicating that heterotrophs took the most advantage of the DDN in this oligotrophic ecosystem.

# 1 Introduction

The availability of nitrogen (N) is one of the key factors controlling primary productivity (PP) in the Ocean (Moore et al., 2013). By supplying new N to surface waters, biological $N_2$-fixation, mediated by some prokaryotes called diazotrophs, plays a critical role in sustaining PP in N-deprived waters such as subtropical gyres (Capone et al., 2005; Karl et al., 2002). The large filamentous bloom-forming cyanobacteria *Trichodesmium* spp. and the diatoms-diazotrophs associations (DDAs) were first thought to be the main contributors to oceanic $N_2$-fixation (Capone et al., 1997; LaRoche and Breitbarth, 2005; Mague et al., 1974). However, the use of molecular tools has demonstrated that the diversity of diazotrophs was greater than previously thought, highlighting in particular the role of pico- and nano-planktonic unicellular cyanobacteria, termed UCYN (Needoba et al., 2007; Zehr et al., 1998, 2001). The latter are now considered to be of a major importance in the global $N_2$ fixation budget due to their broad distribution and high abundance in several oceanic basins (Luo et al., 2012; Moisander et al., 2010; Needoba et al., 2007). These observations are confirmed by the high contribution of $N_2$ fixation rates reported in the $< 10\,\mu m$ size fraction (Bonnet et al., 2009; Dore et al., 2002; Montoya et al., 2004).

While studies dealing with the diversity and the biogeographical distribution of diazotrophs in the ocean are on the increase, little is known regarding the fate of the fixed $N_2$ by the diazotrophs (hereafter called diazotroph-derived N, DDN) in the ocean. It remains unclear whether the DDN is preferentially directly exported out of the photic zone, recycled by the microbial loop, or transferred into larger organisms, subsequently enhancing indirect particle export. Some studies report low $\delta^{15}N$ signatures on zooplankton, evidencing the transfer of DDN towards higher trophic levels (Montoya et al., 2002). This transfer can be direct through the ingestion of diazotrophs (O'Neil et al.,

Discussion Paper | Discussion Paper | Discussion Paper | Discussion Paper |

**BGD**

doi:10.5194/bg-2015-607

**Transfer of diazotroph derived nitrogen**

H. Berthelot et al.

1996; Wannicke et al., 2013), or indirect, i.e. mediated through the release of dissolved N by diazotrophs (Capone et al., 1994; Glibert and Bronk, 1994; Mulholland and Capone, 2001; Mulholland et al., 2004), which is taken up by heterotrophic and autotrophic plankton (Bonnet et al., 2015b), which is subsequently consumed by the zooplankton (e.g. O'Neil et al., 1996). Other studies performed in the tropical North Atlantic and Pacific Oceans report low $\delta^{15}$N signatures on particles from sediment traps, suggesting that at least part of the DDN is ultimately exported out of the photic zone (Karl et al., 2002; Knapp et al., 2005). However, the export efficiency appears to depend on the diazotrophs involved in N$_2$ fixation in surface waters: while it has been demonstrated that DDAs directly contribute to particle export (Karl et al., 2012; Subramaniam et al., 2008; Yeung et al., 2012), *Trichodesmium* spp. is rarely found in sediment traps (Walsby, 1992) mainly due to its positive buoyancy regulated by the production of carbohydrates (Romans et al., 1994). Data on the export efficiency of UCYN are scare. During the VAHINE mesocom experiment designed to track the fate of DDN in the surface oligotrophic ocean, Berthelot et al. (2015b) showed that the production sustained by UCYN (mainly UCYN-C) resulted in a higher rate of particle export compared to the production sustained by DDAs. In this same special issue, Bonnet et al. (2015a) confirmed that UCYN-C significantly contribute to POC export (up to 22.4 ± 5.5 % at the height of the UCYN-C bloom). However, most of the particle export associated with UCYN-C was probably mainly indirect through recycling processes and DDN transfer to surrounding planktonic communities (Bonnet et al., 2015a). However, such transfer of DDN to the surrounding planktonic communities and its potential impact on export production is poorly understood and rarely quantified.

The transfer of DDN to surrounding plankton is mediated through the dissolved pool as diazotrophs release a significant fraction of the fixed N (10–50 %) under the form of ammonium (NH$_4^+$) and dissolved organic N (DON) (Benavides et al., 2013; Glibert and Bronk, 1994; Konno et al., 2010; Mulholland and Bernhardt, 2005; Mulholland et al., 2004). This release of DDN by diazotrophs has been linked to exogenous processes such as viral lysis (Hewson et al., 2004; Ohki, 1999), copepods sloppy feeding

Discussion Paper | Discussion Paper | Discussion Paper | Discussion Paper |

**BGD**

doi:10.5194/bg-2015-607

**Transfer of diazotroph derived nitrogen**

H. Berthelot et al.

Title Page

Abstract | Introduction

Conclusions | References

Tables | Figures

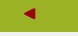 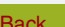

(O'Neil et al., 1996) or programmed cell death (Berman-Frank et al., 2004). Significant N release was also reported in axenic cultures, suggesting that it is also an endogenous process (Mulholland et al., 2004). Once released, fixed N compounds are potentially transferred to non-diazotrophic plankton communities, as suggested by massive developments of diatoms (Devassy et al., 1979; Dore et al., 2008; Lee Chen et al., 2011) and dinoflagellates (Lenes and Heil, 2010; Mulholland et al., 2006) during or following blooms of *Trichodesmium* spp. $^{15}$N-enrichment measured in size fractioned pico-plankton after $^{15}$N$_2$ incubations also supports the idea of a DDN transfer within the planktonic community) (Bryceson and Fay, 1981; Garcia et al., 2007). However, this method probably overestimates the DDN transfer as it is not possible to discriminate between DDN that has been transferred to pico-plankton and N$_2$ fixation by pico-plankton itself. Bonnet et al. (2015b) recently measured the actual transfer of DDN from several *Trichodesmium* spp. blooms to different groups of autotrophic and heterotrophic plankton using single cell mass spectrometry analyses (nanoSIMS) coupled with cell sorting by flow cytometry after $^{15}$N$_2$ labeling, and showed that the DDN was predominantly transferred to diatoms and bacteria, and DDN was mainly converted to diatom biomass. This study was performed during naturally-occuring *Trichodesmium* spp. blooms, but comparative studies on the transfer efficiency of DDN from different diazotrophs are lacking. *Trichodesmium* spp. and UCYN exhibit distinct patterns of N$_2$ fixation (the first fix during the day, while the second fix during the night, e.g. Bergman et al., 2013; Dron et al., 2012) and inhabit different ecological niches (Luo et al., 2012), thus having a potentially different fates in the marine food webs, that remains to be explored.

Here, we compared N$_2$ fixation rates, the quantity and the quality of DDN released in the dissolved pool and the transfer of DDN towards non-diazotrophic plankton from three distinct diazotrophic groups: *Trichodesmium erythraeum*, *Crocosphaera watsonii* and *Cyanothece* sp. For this purpose, we simulated blooms of these three diazotroph phylotypes by inoculating freshly sampled seawater containing the natural planktonic assemblage with the three diazotrophic strains grown in culture mimmicing the natural

**BGD**

doi:10.5194/bg-2015-607

**Transfer of diazotroph derived nitrogen**

H. Berthelot et al.

Discussion Paper | Discussion Paper | Discussion Paper | Discussion Paper

environment. NanoSIMS was used in combination with flow cytometry cell sorting and $^{15}N_2$ labelling to trace the passage of $^{15}$N-labelled DDN into several groups of non-diazotrophic phytoplankton and bacteria to compare the DDN transfer efficiency from these three diazotroph groups.

## 2 Material and methods

### 2.1 Experimental setup

This experiment was carried in the New Caledonian lagoon (southwestern Pacific), which is a tropical low-nutrient low-chlorophyll (LNLC) system. The specific location at the entrance of the lagoon, 28 km off the coast (166.44° E, 22.48° S) was selected as this was the site where the 23 day VAHINE mesocosm experiment presented in this current issue was implemented in the austral summer of 2013. The VAHINE experiment was designed to track the fate of DDN in the ecosystem during a UCYN-C bloom (Bonnet et al., 2015c). The present experiment performed in microcosms was designed to complement the mesocosm experiment and compare the fate of DDN originating from distinct groups of diazotrophs.

### 2.1.1 Cultures maintenance

Three unialgal cultures of diazotrophs abundant in the southeastern Pacific (e.g. Bonnet et al., 2015d; Turk-Kubo et al., 2015) were used in this study to simulate blooms of the filamentous colony forming *Trichodesmium erythraeum* IMS101, and the UCYN strains *Crocosphaera watsonii* WH8501 and *Cyanothece* ATTC51142. They were grown in batch cultures under close to lagoon conditions, and maintained in exponentially growing phase under $120$ photons m$^{-2}$ irradiance on a 12 : 12 light : dark cycle at 27 °C. The culture medium was composed of 0.2 μm filtered and sterilized seawater collected in the New Caledonian lagoon (166.44° E, 22.48° S), at the study site where the DDN transfer experiment described below was performed. The collected seawater

**BGD**

doi:10.5194/bg-2015-607

**Transfer of diazotroph derived nitrogen**

H. Berthelot et al.

**BGD**

doi:10.5194/bg-2015-607

**Transfer of diazotroph derived nitrogen**

H. Berthelot et al.

was characterized by low nitrate + nitrite ($NO_x$) concentrations ($< 0.1\,\mu mol\,L^{-1}$). It was amended with phosphate ($PO_4^{3-}$) and micronutrients according to the N-deplete YBCII medium recipe (Chen et al., 1996), except for $PO_4^{3-}$ concentration, which was reduced to $10\,\mu mol\,L^{-1}$ instead of $50\,\mu mol\,L^{-1}$ in the original medium. Cultures were acclimated
to this medium for at least 10 generations before the experiment started. They were not axenic but manipulations under laminar flow hood and sterilization of the lab materials were performed in order to limit bacterial contamination. Before inoculation into natural seawater, and in order to control the biomass of diazotrophs added, cultures were monitored microscopically every 1–2 days on a Malassez counting cell for UCYN and
on a 10 µm polycarbonate filter for *T. erythraeum*, using an epifluorescence microscope (Zeiss Axioplan, Jana, Germany) fitted with a green (510–560 nm) excitation filter.

## 2.1.2  DDN transfer experiment

Seawater containing the natural planktonic community was collected at the experimental study site on 2 February 2014 at 2 m depth, using an air-compressed Teflon
pump (AstiPure™) connected to a polyethylene tubing. At the time of the sampling, the seawater temperature was 25.4 °C. Ambient $PO_4^{3-}$ and $NO_x$ concentrations were $< 0.2\,\mu mol\,L^{-1}$. Seawater was transferred into 15 HCl-washed 4.5 L polycarbonate bottles equipped with septum caps and quickly brought back to the laboratory. Bottles were divided into five sets of three replicates. The first set was immediately amended
with *Trichodesmium erythraeum* (hereafter referred to as "*T. erythraeum* treatment"), the second with the UCYN *Crocosphaera watsonii* (hereafter referred to as "*C. watsonii* treatment"), the third one with the UCYN *Cyanothece* spp. (hereafter referred to as "*Cyanothece* treatment"), the fourth set was left unamended and served as a control (hereafter referred to as "Control treatment"), and the last set was immediately
processed as described below to characterize the initial conditions (T0). To simulate blooms of the different diazotrophs, we added $5.10^3$ trichomes $L^{-1}$ for *T. erythraeum* treatment and $1.10^6$ cells $L^{-1}$ for the UCYN treatments, to be representative of the di-

azotroph blooms observed in the South West Pacific region (Bonnet et al., 2015d; Moisander et al., 2010; Rodier and Le Borgne, 2008; Shiozaki et al., 2014). Care was made to introduce a similar biomass of diazotrophs in each treatments in order to be able to compare the different treatments. The initial cultures were sufficiently concentrated in cells in such a way that the volume of culture added represented less than 1 % of the 4.5 L bottles volume, so nutrient concentrations, especially $PO_4^{3-}$ concentrations were not significantly influenced by these additions, which represented $< 0.05\,\mu mol\,L^{-1}$ of added $PO_4^{3-}$.

Immediately after the diazotrophs inoculation, all 4.5 L bottles were amended with $NaH^{13}CO_3$ (EURISOTOP, 99 atom % $^{15}$N, 0.37 g in 60 mL of deionized water) to obtain a $\sim 10$ atom % $^{13}$C-enrichment (1 mL in each 4.5 L bottles) and $^{15}N_2$ (98.9 atom % $^{15}$N, Cambridge isotopes) enriched seawater, according to the protocol developed by Mohr et al. (2010) and fully described in Berthelot et al. (2015). Briefly, $^{15}N_2$ enriched seawater was prepared by circulating 0.2 µm filtered seawater collected at the same site as described above through a degassing membrane (Membrana, Minimodule®, flow rate 450 mL min$^{-1}$) connected to a vacuum pump ($< 850$ mbar) for at least 1 h. The degassed seawater was transferred to a 2 L gas tight Tedlar® bag and amended with 1 mL of $^{15}N_2$ per 100 mL of seawater. The $^{15}N_2$ bubble was vigorously shaken until its complete dissolution. The incubation bottles were then amended with 5 % vol : vol enriched seawater and closed without headspace with septum caps. The final $^{15}$N-enrichment of the $N_2$ pool in the incubation bottles was measured using a Membrane Inlet Mass Spectrometer (Kana et al., 1994) and was found to be $3.5 \pm 0.2$ atom % ($n = 9$).

Except for the T0 set of bottles, all bottles were incubated for 48 h under in situ-simulated conditions in on-deck incubators at $\sim 26.5\,^{\circ}$C with continuous water flowing at temperature and irradiances corresponding to the sampling depth using neutral screening. Bottles were gently mixed three times per day during the experiment to insure homogeneity. After incubation, the four sets of bottles (the three diazotrophs-amended treatments and the control treatment) were recovered and sub-sampled to analyze the following parameters: heterotrophic bacteria and phytoplankton abundances, $N_2$ fixa-

Discussion Paper | Discussion Paper | Discussion Paper | Discussion Paper |

**BGD**

doi:10.5194/bg-2015-607

**Transfer of diazotroph derived nitrogen**

H. Berthelot et al.

**BGD**

doi:10.5194/bg-2015-607

**Transfer of diazotroph derived nitrogen**

H. Berthelot et al.

tion rates, DDN release, organic and inorganic nutrients concentrations and cellular [15]N- and [13]C-enrichment on diazotrophs and non-diazotrophic plankton groups (see below for detailed protocols). Unless otherwise stated, samples were taken individually in each bottle of each set, so each parameter was measured in triplicate in every

treatment.

## 2.2  Plankton abundance determination

Samples for micro-phytoplankton were collected from the 4.5 L incubation bottles in 250 mL glass bottles and fixed with lugol (0.5 % final concentration). Diatoms, dinoflagellates and micro-zooplankton (ciliates) were identified and enumerated to the lowest

possible taxonomic level from a 100 mL subsample following the Utermohl methodology (Hasle, 1978), using a Nikon Eclipse TE2000-E inverted microscope equipped with phase-contrast and a long distance condenser.

Pico-, nano-phytoplankton and bacterial abundances were determined using flow cytometry. For this purpose, samples were collected in 1.8 mL cryotubes, fixed

with paraformaldehyde (final concentration 2 %), left at ambient temperature for 15 min in the dark, flash frozen in liquid $N_2$ and stored at $-80\,°C$. Analyses were carried out at the PRECYM flow cytometry platform (https://precym.mio.univ-amu.fr/) using standard flow cytometry protocols (Marie et al., 1999) to enumerate phytoplankton and heterotrophe bacteria, using a FACSCalibur analyzer (BD Biosciences, San Jose, CA).

Samples were thawed at room temperature and just before analyses, were added to each sample: 2 μm beads (Fluoresbrite YG, Polyscience), used as internal control (and to discriminate picoplankton < 2 μm < nanoplankton populations), and Trucount beads (BD Biosciences), used to determine the volume analyzed. An estimation of the flow rate was calculated weighing 3 tubes of samples before and after a 3 mn run of the

cytometer. The cell concentration was determined from both Trucount beads and flow rate measurements. For picoplankton cells, the red fluorescence (670LP, related to chlorophyll *a* content) was used as trigger signal and cells were characterized by 3 other optical signals: forward scatter (FSC, related to cell size), side scatter (SSC, re-

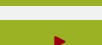
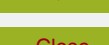

Discussion Paper | Discussion Paper | Discussion Paper | Discussion Paper | Discussion Paper |

**BGD**

doi:10.5194/bg-2015-607

**Transfer of diazotroph derived nitrogen**

H. Berthelot et al.

lated to cell structure), and the orange fluorescence (580/30 nm, related to phycoery-thrin content). Phytoplankton communities were clustered as *Synechococcus* spp. cell like (hereafter called *Synechococcus*), *Prochlorococcus* spp. cell like (hereafter called *Prochlorococcus*) and pico- and nano-eukaryotes ($< 20$ µm, hereafter referred to as small eukaryotes). In addition, in the UCYN treatments, *C. watsonii* and *Cyanothece* clusters were determined. The resolution of these clusters was realized by comparing the UCYN treatments cytograms with the control one. The proportion of diazotrophic cells in these clusters (i.e. the proportion of the new counts in the UCYN treatments compared to the control treatment) was $> 98$ and $> 90$ % for *C. watsonii* and *Cyanothece*, respectively. For heterotrophic bacteria (hereafter called "bacteria") samples were stained with SYBR Green II (Molecular Probes, final conc. 0.05 % [$v/v$], for 15 min at room temperature in the dark), in order to stain nucleic acids then cells were charac-terized by 2 main optical signals: side scatter (SSC, related to cell size and structure) and green fluorescence (530/40, related to SYBR Green fluorescence). For the calcu-lation of heterotrophic prokaryotes abundances, phytoplankton cells, *Prochlorococcus* and *Synechococcus* particularly, were gated out on the basis of their chlrorphyll *a* con-tent (red fluorescence) (Sieracki et al., 1995). All data were collected in log scale and stored in list mode using the CellQuest software (BD Biosciences). Data analysis was performed a posteriori using SUMMIT v4.3 software (Dako).

The abundance of *T. erythraeum* added to the natural planktonic assemblage was monitored microscopically: 300 mL from the 4.5 L bottles were filtered on a 10 µm poly-carbonate filter in each triplicate bottle. The cells were fixed with paraformaldehyde (2 % final concentration) for at least 1 h at 4 °C and stored at −20 °C until counting us-ing an epifluorescence microscope (Zeiss Axioplan, Jana, Germany) fitted with a green (510–560 nm) excitation filter.

## 2.3   N₂ fixation rates determination

For net N₂ fixation, 2 L from each 4.5 L bottle were filtered onto precombusted (450 °C, 4 h) GF/F filters. Filters were stored at −20 °C and dried at 60 °C for 24 h before analy-

Discussion Paper | Discussion Paper | Discussion Paper | Discussion Paper |

**BGD**

doi:10.5194/bg-2015-607

**Transfer of diazotroph derived nitrogen**

H. Berthelot et al.

sis. The particulate organic N (PON) content and PON $^{15}$N isotopic enrichment of each filter were measured by continuous-flow isotope ratio mass spectrometry coupled to an elemental analyser (EA-IRMS) using an Integra-CN mass spectrometer. The analytical precision associated with the mass determination averaged 2.8 % for PON. The analytical precision associated with $^{15}$N was ±0.0010 atom % $^{15}$N for a measured mass of 0.7 µmol N. The particulate inorganic N contribution was not taken into account. $N_2$ fixation rates were calculated according to Montoya et al. (1996). We considered the results to be significant when $^{15}$N excess enrichment was higher than three times the standard deviation obtained with time zero samples ($n = 3$).

## 2.4 DDN released to the dissolved pool

300 mL of the filtrate obtained during $N_2$ fixation filtrations was recovered and stored in 500 mL SCHOTT glass flasks, poisoned with $HgCl_2$ (final concentration 10 µg L$^{-1}$) and stored at 4 °C for further measurement of the $^{15}$N-enrichment of the dissolved pool. This was achieved using the two step diffusion method extensively described in Berthelot et al. (2015a) and derived from Slawyk and Raimbault (1995). This method enables the differentiation of the $NH_4^+$ and DON pools and measures their respective $^{15}$N-enrichment. It should be noted that in the DON recovery step, $NO_x$ were also recovered. However, $NO_x$ concentrations were very low during our experiments ($< 0.2$ µmol L$^{-1}$) with respect to DON concentrations ($\sim 4.5$ µmol L$^{-1}$). Furthermore, they were unlikely to be released by diazotrophs, thus unlikely $^{15}$N-enriched, and their contribution was not taken into account. Net DDN release rates were calculated according to Berthelot et al. (2015a).

## 2.5 Organic and inorganic nutrient analyses

Samples for $NH_4^+$ concentrations determination were collected in duplicate in 40 mL SHOTT flasks and $NH_4^+$ concentrations were measured according to Holmes

et al. (1999) using a trilogy fluorometer (Turner Design, quantification limit = 3 nmol L$^{-1}$). Samples for inorganic nutrients were collected in triplicate in 20 mL acid washed scintillation vials, poisoned with HgCl$_2$ (10 µg L$^{-1}$ final concentration) and stored in the dark at 4 °C until analysis. NO$_x$ and PO$_4^{3-}$ concentrations were determined by standard colorimetric procedures (Aminot and Kérouel, 2007) on a segmented flow auto-analyzer. The quantification limit was 0.05 µmol L$^{-1}$. Samples for determination of DON concentrations were collected in 40 mL SHOTT flasks after filtration onto combusted GF/F filters (450 °C, 4 h) and stored at −20 °C until analysis. Concentrations were measured by wet oxidation according to Pujo-Pay and Raimbault (1994).

## 2.6 Cell sorting and sampling for nanoSIMS analyses

For flow cytometry cell sorting and subsequent analysis using nanoSIMS, samples were collected as follows to pre-concentrate cells and facilitate cell sorting: for each treatment, 300 mL of each triplicate from the 4.5 L bottle were pooled and filtered onto 0.2 µm pore size 47 mm polycarbonate filters. Filters were quickly placed in a 5 mL cryotube® filled with 0.2 µm filtered seawater with PFA (2 % final concentration), for at least 1 h at room temperature in the dark.The cryovials were vortexed, for at least 10 s, in order to detach the cells from the filter and were stored at −80 °C until analysis. Cell sorting was performed on a Becton Dickinson Influx$^{TM}$ Mariner (BD Biosciences, Franklin Lakes, NJ) high speed cell sorter of the Regional Flow Cytometry Platform for Microbiology (PRECYM), hosted by the Mediterranean Institute of Oceanography, as described in Bonnet et al. (2015b). Planktonic groups were separated using the same clusters as for the phytoplankton abundance determination as described above. After sorting, the cells were recovered in ependorf tubes and immediately filtered onto a 0.2 µm pore size 25 mm filter. Particular care was taken to drop the cells on the surface as small as possible (∼ 5 mm in diameter) to ensure the highest cell density possible to facilitate further nanoSIMS analyses. In the UCYN treatments, additional

**BGD**

doi:10.5194/bg-2015-607

**Transfer of diazotroph derived nitrogen**

H. Berthelot et al.

"diazotroph" sort gates were defined. The gates were delimited around the new populations that appeared in the UCYN treatments, compared to the control.

Large phytoplanktonic cells (*T. erythraeum* and diatoms) were visible and easily recognized on the CCD camera of the nanoSIMS and thus did not require any cell sorting step. Thus, to recover these cells, 300 mL of each triplicate 4.5 L bottle were pooled together and filtered on 10 µm pore size 25 mm polycarbonate filters. The cells were fixed with PFA (2 % final concentration) for at least 1 h at ambient temperature. The filters were then stored at −20 °C until nanoSIMS analyses.

## 2.7 NanoSIMS analyses and data processing

NanoSIMS analyses were performed using a NanoSIMS N50 at the French National Ion MicroProbe Facility according to Musat et al. (2008) and Bonnet et al. (2015b). Briefly, a ∼ 1.3 pA Cesium (16 KeV) primary beam focused onto ∼ 100 nm spot diameter was scanned across a 256 × 256 or 512 × 512 pixel raster (depending on the image size) with a counting time of 1 ms per pixel. Samples were pre-sputtered prior to analyses to achieve sputtering equilibrium. Negative secondary ions ($^{12}C^{-}$, $^{13}C^{-}$, $^{12}C^{14}N^{-}$, $^{12}C^{15}N^{-}$, and $^{28}Si^{-}$) were collected by electron multiplier detectors, and secondary electrons were also imaged simultaneously. Ten to fifty serial quantitative secondary ion images were generated, that were combined to create the final image. Mass resolving power was ∼ 8000 in order to resolve isobaric interferences. From 20 to 100 planes were generated for each cells analyzed. NanoSIMS runs are time-intensive and not designed for routine analysis, but at least 20 cells from each community were analysed to assess the variability in isotopic composition under the same conditions. Thus, for diatoms only the three dominant species present in our experiment and previously counted microscopically were analysed. Data were processed using the LIMAGE and Look@NanoSIMS (Polerecky et al., 2012) software. Briefly, all scans were corrected for any drift of the beam and sample stage during acquisition. Isotope ratio images were created by adding the secondary ion counts for each recorded secondary ion for each pixel over all recorded planes and dividing the total counts by the total counts of

Discussion Paper | Discussion Paper | Discussion Paper | Discussion Paper |

**BGD**

doi:10.5194/bg-2015-607

**Transfer of diazotroph derived nitrogen**

H. Berthelot et al.

a selected reference mass. Individual cells were easily identified in nanoSIMS $^{12}$C, $^{14}$N and $^{28}$Si images that were used to define regions of interest (ROIs) around individual cells. For each ROI, the $^{15}$N and $^{13}$C enrichment were calculated. In total, almost 1000 ROIs were used for this study.

## 2.8 Cell-specific biomass and DDN transfer calculations

The biomass of the added diazotrophs was measured at T0 by filtering an aliquote of each culture on a precombusted GF/F filter for PON determination as described above. The total biomass was divided by the number of cells determined microscopically to obtain the cell-specific biomass.

For diatoms, the biovolume of the three most abundant diatom taxa (*Chaetoceros* spp., *Bacteriastrum* spp. and *Thalassionema nitzschioides*) was estimated by measuring their cross, apical and transapical sections in order to calculate their biovolume according to Sun and Liu (2003). At least 50 measurements were performed for each diatom taxon. Biovolume was then converted to N cellular content according to Smayda et al. (1978) and using a C : N ratio of 6.6 : 1 (Redfield, 1934). These three taxa represented $\sim 75\%$ of the total diatom abundance in this experiment. The remaining 25 % was mainly composed of smaller diatoms (e.g. *Pseudo-Nitzschia* spp., *Cylindrotheca* spp. and *Leptocylindrus* spp.) that probably weakly contributed to the total diatom biomass.

For *Synechococcus*, the C content reported in Buitenhuis et al. (2012) was used ($255\,\mathrm{fg\,C\,cell^{-1}}$) and converted into N content according to the Redfield ratio of 6.6 : 1 leading to a value of $3.2 \pm 0.9\,\mathrm{fmol\,N\,cell^{-1}}$. For bacteria, the average N content of $0.15 \pm 0.08\,\mathrm{fmol\,N\,cell^{-1}}$ (Fukuda et al., 1998) was assumed. For the small eukaryotes, the cellular N content of $9.2 \pm 2.9\,\mathrm{fmol\,cell^{-1}}$ was used as reported in Gregori et al. (2001). The cellular N content of each group multiplied by their abundances allowed the calculation of the biomasses associated with each plankton group.

**BGD**

doi:10.5194/bg-2015-607

**Transfer of diazotroph derived nitrogen**

H. Berthelot et al.

The DD$^{15}$N transfer (in nmol L$^{-1}$ 48 h$^{-1}$) that depict the amount of $^{15}$N$_2$ transferred from diazotrophs towards the non-diazotrophic plankton was calculated for each plankton group analysed as follows:

$$DD^{15}N = \frac{R_{cell}}{R_{N_2}} \times N_{con} \times A \qquad (1)$$

where $R_{cell}$ is the mean $^{15}$N-enrichment of individual cells (in atom %) after 48 h of incubation, $R_{N_2}$ is the $^{15}$N-enrichment of the $^{15}$N$_2$ in the dissolved pool (in atom %), $N_{con}$ is the cellular N content (in nmol N cell$^{-1}$) and $A$ is the plankton group specific abundance (in cell L$^{-1}$).

## 2.9 Statistical analyses

The effect of the diazotrophs treatments on the biomass associated with non-diazotrophs was tested using an Tukey HSD test. The differences in the $^{15}$N-enrichment of cells between the different treatments and the natural abundance were tested using an unpaired non-parametric Mann–Whitney test, as the dispersion of values did not follow a normal distribution pattern. The statistical significance threshold was 5 % ($p < 0.05$). All the uncertainties associated with the parameters measured were taken into account and propagated over the different computations made.

## 3 Results

### 3.1 Plankton abundance and biomass

At the start of the experiment (T0), (i.e. ambiant waters in which the DDN tranfer experiment was performed), diatoms dominated the micro-phytoplanktonic community (89 % of the total abundance), mainly driven by the contribution of *Chaetoceros* spp. (6130 cells L$^{-1}$), *Thalassionema* spp. (5345 cells L$^{-1}$) and *Bacteriastrum*

Discussion Paper | Discussion Paper | Discussion Paper | Discussion Paper |

**BGD**

doi:10.5194/bg-2015-607

**Transfer of diazotroph derived nitrogen**

H. Berthelot et al.



spp. (2391 cells L$^{-1}$), which together represented $\sim$ 75 % of the total diatom community (Table S1 in the Supplement). Dinoflagellates were an order of magnitude less abundant than diatoms and were mainly composed of *Gymnodinium* spp. and *Gyrodinium* spp. Few *Trichodesmium* spp. filaments were observed in the natural assemblage at abundances lower than 40 trichomes L$^{-1}$. Ciliate abundance was 430 cells L$^{-1}$ including 40 to 100 tintinnids cells L$^{-1}$. The initial abundance of *Synechococcus*, *Prochlorococcus*, small eukaryotes and bacteria determined by flow cytometry was $5.4 \pm 1.1 \times 10^4$, $2.2 \pm 0.4 \times 10^4$, $1.4 \pm 0.1 \times 10^3$ and $5.9 \pm 1.5 \times 10^5$ cells mL$^{-1}$, respectively (Table S1).

Converted to biomass, *Synechococcus* dominated to phytoplanktnic biomass at T0 (120 $\pm$ 40 nmol N L$^{-1}$), followed by bacteria (90 $\pm$ 40 nmol N L$^{-1}$) and diatoms (40 $\pm$ 14 nmol N L$^{-1}$). The biomass associated with small eukaryotes and *Prochlorococcus* together represented less than 10 nmol L$^{-1}$ (i.e. 3 % of the total biomass). The dinoflagellate and ciliate biomass was one to two orders of magnitude lower than the diatom biomass, respectively, and were thus not considered in detail in this study.

In the control treatment after 48 h of incubation, the abundance of total diatoms and dinoflagellates increased by a factor of 2.3 and 1.9, respectively, while the abundances of bacteria remained stable and *Synechococcus* and *Prochlorococcus* abundances decreased by a factor of 1.4 and 1.3 respectively (Table S1). In the diazotrophs-amended treatments, the abundance of added diazotrophs decreased slightly in the *T. erythraeum* treatment (from $5 \times 10^3$ to $3.9 \pm 0.5 \times 10^3$ trichomes L$^{-1}$) and remained stable around $1 \times 10^6$ cells L$^{-1}$ in the UCYN treatments (Table S1).

After 48 h of incubation, the biomass associated with non-diazotrophs increased in all the diazotrophs-amended treatments compared to the control (Fig. 1). The highest increase was observed in the *T. erythraeum* treatment (62 $\pm$ 39 %), mainly driven by a bacterial biomass increase of 90 $\pm$ 6 % and to a lesser extent by a *Synechococcus* (47 $\pm$ 22 %) and diatom (37 $\pm$ 17 %) biomass increase (Figs. 1 and 2). In the *C. watsonii* and *Cyanothece* treatments, the increase of biomass associated with non-diazotrophic plankton was 39 $\pm$ 39 and 35 $\pm$ 46 %, respectively. It was mainly driven by bacterial

Discussion Paper | Discussion Paper | Discussion Paper | Discussion Paper | Discussion Paper |

**BGD**

doi:10.5194/bg-2015-607

**Transfer of diazotroph derived nitrogen**

H. Berthelot et al.

(58 ± 12 %), *Synechococcus* (23 ± 10 %) and diatom (30 ± 16 %) biomass increase in the *C. watsonii* treatment, and by bacterial biomass increase only (116 ± 16 %) in the *Cyanothece* treatment. The effect of diazotrophs on the biomass of small eukaryots was less noticeable.

In all the treatments, the sum of the N biomass associated with every group of plankton was in good agreement with the actual PON concentrations measured by EA-IRMS after 48 h, indicating that the cellular N contents used in this study (described in Sect. 2) are realistic (Fig. 2).

## 3.2   $N_2$ fixation rates and DDN release

Net $N_2$ fixation rates determined by EA-IRMS in the control treatment were 1.5 ± 0.1 nmol L$^{-1}$ 48 h$^{-1}$ (Fig. 3). This $N_2$ fixation was attributed to the diazotrophs already present in the natural assemblage (probably *Trichodesmium* spp. that were found at low abundances in the control, data not shown). In the diazotroph-amended treatments, net $N_2$ fixation rates were 10 to 40 times higher than in the control, indicating

the that diazotroph blooms artificially induced worked well: ∼ 60 nmol L$^{-1}$ 48 h$^{-1}$ in the *T. erythraeum* and *C. watsonii* treatments and 16 nmol L$^{-1}$ 48 h$^{-1}$ in the *Cyanothece* treatment (Fig. 3). The DDN released to the dissolved pool by diazotrophs represented 16.1 ± 6.7 % of the total $N_2$ fixation (where total $N_2$ fixation is defined as the sum of $N_2$ fixed recovered in the PON, DON and $NH_4^+$ pools) in the *T. erythraeum* treatment,

13.8 ± 1.9 % in the *C. watsonii* treatment, 30.5 ± 10.4 % in the *Cyanothece* treatment and 66.0 ± 21.9 % in the control treatment. In all cases, most of the $^{15}$N released in the dissolved pool after 48 h of incubation was under the form of DON, which represented 77 to 81 % of the total N release in the diazotrophs-amended treatments without any differences between the treatments. The $NH_4^+$ release was below detection limit in the

control treatment.

**BGD**

doi:10.5194/bg-2015-607

**Transfer of diazotroph derived nitrogen**

H. Berthelot et al.

Discussion Paper | Discussion Paper | Discussion Paper | Discussion Paper |

### 3.3 Cell-specific $^{15}$N- and $^{13}$C-enrichments and DD$^{15}$N transfer towards the non-diazotrophic plankton

NanoSIMS analyses revealed significant $^{15}$N-enrichment in diazotrophic cells after 48 h of incubation compared to natural $^{15}$N-enrichment (0.366 atom %) (Figs. 4 and 5). Among the three diazotrophs added, *C. watsonii* and *Cyanothece* exhibited the highest $^{15}$N-enrichments with 1.942±0.239 atom % ($n = 18$) and 2.501±0.300 atom % ($n = 46$), respectively (Fig. 5). *T. erythraeum* $^{15}$N-enrichment averaged 1.147 ± 0.233 atom % ($n = 68$). The $^{13}$C-enrichment was similar for *T. erythraeum* (3.316±0.634 atom %) and *C. watsonii* (3.124 ± 0.670 atom %) and higher for *Cyanothece* (4.612 ± 0.837 atom %). The correlation between $^{13}$C-enrichment and $^{15}$N-enrichment was significant for *T. erythraeum* ($r^2 = 0.50$, $p < 0.001$, $n = 68$), weaker but still significant for *C. watsonii* ($r^2 = 0.39$, $p = 0.005$, $n = 18$), and not significant for *Cyanothece* ($r^2 = 0.01$, $p = 0.500$, $n = 46$).

Cell specific $N_2$ fixation rates of diazotrophs were 140.8 ± 55.9 fmol N cell$^{-1}$ 48 h$^{-1}$ (assuming 100 cell per trichomes), 50.3 ± 9.2 fmol N cell$^{-1}$ and 25.0 ± 3.5 fmol N cell$^{-1}$ 48 h$^{-1}$ for *T. erythraeum*, *C. watsonii* and *Cyanothece* leading to $N_2$ fixation rates associated with the three groups of 54.8 ± 21.7, 54.5 ± 10.0 and 19.1 ± 2.7 nmol L$^{-1}$ 48 h$^{-1}$, respectively.

NanoSIMS analyses performed on non-diazotrophic diatoms and cell-sorted *Synechococcus*, small eukaryotes, and bacteria also revealed $^{15}$N-enrichments that were at times significantly higher than those measured in the control (Figs. 4 and 6). The $^{15}$N-enrichment of non-diazotrophic plankton strongly depended on the treatment considered. When *T. erythraeum* provided the DD$^{15}$N, the $^{15}$N-enrichment was significantly higher compared to the control for diatoms (0.468 ± 0.081 atom %, $n = 18$), *Synechococcus* (0.404 ± 0.090 atom %, $n = 105$) and bacteria (0.487 ± 0.071 atom %, $n = 45$) (data are not available for small eukaryotes in *T. erythraeum* treatment). In the *C. watsonii* treatment, the $^{15}$N-enrichment of non-diazotrophs was significantly higher compared to the control for *Synechococcus* (0.411 ± 0.079 atom %, $n = 134$)

Discussion Paper | Discussion Paper | Discussion Paper | Discussion Paper |

**BGD**

doi:10.5194/bg-2015-607

**Transfer of diazotroph derived nitrogen**

H. Berthelot et al.

and bacteria ($0.435 \pm 0.05$ atom %, $n = 34$) and not significantly different for diatoms ($0.394 \pm 0.077$ atom %, $n = 23$) and small eukaryotes ($0.383 \pm 0.040$ atom %, $n = 52$). In the *Cyanothece* treatment, the $^{15}$N-enrichment of non-diazotrophs was significantly higher compared to the control for diatoms ($0.446 \pm 0.143$ atom %, $n = 26$) and for *Synechococcus* ($0.389 \pm 0.080$ atom %, $n = 25$), whereas no significant enrichments were observed for small eukaryotes ($0.383 \pm 0.030$ atom %, $n = 88$) and bacteria ($0.379 \pm 0.027$ atom %, $n = 38$). It should be noted that in the control, the $^{15}$N-enrichment of all plankton groups (diatoms, *Synechococcus*, small eukaryotes and bacteria) was slightly higher ($0.387 \pm 0.048$ atom %, $n = 301$) than the natural abundance ($0.366$ atom %) after 48 h of incubation (Fig. 6).

The amount of DD$^{15}$N transferred to non-diazotrophs corrected from N$_2$ fixation detected in the control treatment was higher in the *T. erythraeum* treatment ($9.5 \pm 4.9$ nmol N L$^{-1}$) compared to the *C. watsonii* and *Cyanothece* treatments, where it was $4.1 \pm 2.3$ and $1.2 \pm 0.9$ nmol N L$^{-1}$, respectively. It represented $11.7 \pm 4.4$ % of total N$_2$ fixation in the *T. erythraeum* treatment and was significantly higher than in the *C. watsonii* ($5.8 \pm 2.7$ %) and *Cyanothece* treatments ($4.9 \pm 2.4$ %) (Table 1).

## 4  Discussion

While DDN is the major source of external N for the surface ocean (Gruber, 2004), its fate in the marine food web has been poorly studied, mainly due to technical limitations. Furthermore, the coexistence of a wide diversity of diazotrophs in the surface ocean (e.g. Moisander et al., 2010) raises the question of whether or not the fate of DDN depends on the diazotroph phylotypes involved in N$_2$ fixation. Using $^{15}$N and $^{13}$C labeling coupled with cell sorting by flow cytometry and nanoSIMS analyses at the single cell level, we were able to trace the transfer of DD$^{15}$N from the diazotrophs to the dissolved pool and to the non-diazotrophic plankton, and compare the DD$^{15}$N transfer efficiency as a function of the diazotroph groups dominating the community.

**BGD**

doi:10.5194/bg-2015-607

**Transfer of diazotroph derived nitrogen**

H. Berthelot et al.

**BGD**

doi:10.5194/bg-2015-607

**Transfer of diazotroph derived nitrogen**

H. Berthelot et al.

Discussion Paper | Discussion Paper | Discussion Paper | Discussion Paper

## 4.1 Cell-specific photosynthesis and $N_2$ fixation

Cell-specific $N_2$ fixation rates measured using nanoSIMS are in the range of previous $N_2$ fixation rates measured in cultures using conventional $N_2$ fixation methods for the same strains of *T. erythraeum, C. watsonii* and *Cyanothece* (Berthelot et al., 2015a). This confirms the ability of nanoSIMS to accurately measure $N_2$ fixation rates, as previously shown in former studies (Finzi-Hart et al., 2009; Foster et al., 2013; Ploug et al., 2010). The high $N_2$ fixation rates induced by the inoculation of diazotrophs in the natural planktonic community ($7–30\,\mathrm{nmol\,N\,L^{-1}\,d^{-1}}$) are representative of those reported in the South West Pacific region under blooming conditions (Berthelot et al., 2015b; Bonnet et al., 2015d; Garcia et al., 2007). Thus, the artificial diazotroph blooms induced for the purpose of this study provided realistic conditions to study the DDN transfer to non-diazotrophic plankton.

The significant correlation between $^{13}$C- and $^{15}$N-enrichments in *T. erythraeum* cells analyzed after 48 h of incubation argue that both PP and $N_2$ fixation occur simultaneously within the cells (Fig. 5). This appears in opposition with the idea of the cells specialization in $N_2$ fixation (called diazocytes) where high respiration rates and degradation of glycogen and gas vacuoles reduce the $O_2$ concentration enabling the expression of *nif* genes allowing daytime $N_2$ fixation (Bergman and Carpenter, 1991; Berman-Frank et al., 2001; Sandh et al., 2012). However, it has to be noticed that after 48 h of incubation with the tracers, it is highly probable that both $^{15}$N and $^{13}$C have been exchanged between cells, leading to a homogenization of the cells isotopic enrichments.

More surprisingly, the coupling between $^{13}$C- and $^{15}$N-enrichments for individual UCYN cells after 48 h of incubation is weaker than for *T. erythraeum* cells, in particular for *Cyanothece*. This appears counter-intuitive as UCYN are supposed to perform both $N_2$ fixation and photosynthesis within the same cell. This uncoupling suggests that UCYN cells might be at least partially specialized in photosynthesis or $N_2$ fixation, similarly to *Trichodesmium* spp. These results confirm the patterns already observed

for *C. watsonii* (Foster et al., 2013). In addition, the weaker correlation between [13]C- and [15]N-enrichments in UCYN cells compared to *T. erythraeum* also suggests weaker extracellular fixed N and C exchanges between cells. These differences might be the result of the greater spatial proximity of *Trichodesmium* spp. cells within colonies and

5 filaments compared to free living UCYN cells in the water column. According to this vision, the high production of extracellular polymeric substances observed in different *C. watsonii* strains (Sohm et al., 2011; Webb et al., 2009) might be a strategy to agglomerate the free living UCYN together to form colonies (Bonnet et al., 2015a; Foster et al., 2013), ensuring a spatial proximity and thus facilitating the exchange of metabolites

between cells.

## 4.2  DDN release to the dissolved pool

The DD[15]N released to the dissolved pool after 48 h accounted from 7 to 17 % of total $N_2$ fixation over the three diazotroph-amended treatments. These values are at the lower end of values (10–80 %) reported in *Trichodesmium* spp. blooms in the tropical

Atlantic (Glibert and Bronk, 1994; Mulholland et al., 2006), Southwestern Pacific (Bonnet et al., 2015b) or in mixed diazotroph assemblages of the North Pacific (Konno et al., 2010) and the Atlantic ocean (Benavides et al., 2013). In contrast, these values of N release are at least one order of magnitude higher than those reported in unialgal cultures (< 2 %) for the same strains as those studied here (Berthelot et al., 2015a). It is prob-

able that, in culture, the cells are maintained in optimal growth conditions (exponential growth phase, appropriate light, temperature and nutrient conditions) and optimize the N use, either through a low excretion rate of DDN or through an efficient uptake of DDN (Mulholland et al., 2001). Conversely, in the field, the sampling does not necessarily occur during the exponential growth phase, and exogenous factors may affect the release

of DDN, such as viral lysis (Hewson et al., 2004; Ohki, 1999) and sloppy feeding (O'Neil et al., 1996). In this study, the diazotrophs added to natural seawater were healthy but may have been affected by exogenous factors after innoculation, leading to a moderate

**BGD**

doi:10.5194/bg-2015-607

**Transfer of diazotroph derived nitrogen**

H. Berthelot et al.

Discussion Paper | Discussion Paper | Discussion Paper | Discussion Paper |

proportion (7–17 %) of DDN released in the dissolved pool. These results indicate that the proportion of $N_2$ fixed released in the dissolved compartment both depends on the cell status and on exogenous factors more than the type of diazotrophs involved in $N_2$ fixation, as previously stated by Berthelot et al. (2015a).

## 4.3 DDN transfer efficiency and pathways

*T. erythraeum* transferred $\sim 12$ % of $DD^{15}N$ towards the non-diazotrophic plankton. This is in good agreement with previous estimates by Bonnet et al. (2015b) using the same methodology, who report a $DD^{15}N$ transfer of 7 to 12 % in naturally-occuring *Trichodesmium* spp. blooms. These results confirm that *Trichodesmium* spp. enhances the development of non-diazotrophic plankton, as already suggested by the frequent observations of co-occurrence or succession of *Trichodesmium* spp. and non-diazotrophs, particularly in N depleted environments (Devassy et al., 1979; Dore et al., 2008; Lee Chen et al., 2011; Lenes and Heil, 2010).

The $DD^{15}N$ transfer was half as efficient (4–5 %) in the UCYN treatments compared to the *T. erythraeum* treatment (Table 1). This is in good agreement with the lower increase in plankton biomass associated with non diazotrophs in the UCYN treatments compared to the *T. erythraeum* treatment (Fig. 2). The ecology of UCYN is less characterized than that of *Trichodesmium* spp. and data on their co-occurrence with non-diazotrophic plankton in the ocean are scare. In this issue, Bonnet et al. (2015a) used the single cell approach described here during a natural occurring bloom of UCYN-C (closely related to *Cyanothece* spp.) in the New Caledonian lagoon and measured a $DD^{15}N$ transfer of $21 \pm 4$ % of the total $N_2$ fixation, mainly towards pico-planktonic communities. This bloom co-occurred with a doubling of *Synechococcus* and pico-eukaryotes abundances, as well as an increase of diatoms (Leblanc et al., in prep) and PP (Berthelot et al., 2015b). *Crocosphaera*-like cells observed in association with the diatom *Climacodium* sp. (Carpenter and Janson, 2000) have also been shown to transfer the recently fixed $N_2$ towards the host diatom cell (Foster et al., 2011). All these

**BGD**

doi:10.5194/bg-2015-607

**Transfer of diazotroph derived nitrogen**

H. Berthelot et al.

Discussion Paper | Discussion Paper | Discussion Paper | Discussion Paper

data confirm that UCYN are able to provide DDN to non-diazotrophic plankton and thus promote marine productivity in N depleted areas.

The transfer of DDN towards phytoplankton or bacteria requires the release of N in the dissolved pool. Surprisingly, the total amount of DD$^{15}$N recovered in the dissolved pool was not a good indicator of the DD$^{15}$N transfer efficiency: the highest release of DDN was measured in the *Cyanothece* treatment ($16.6 \pm 4.9$ % of the total $N_2$ fixation) and led to the lowest DD$^{15}$N transfer efficiency ($4.9 \pm 2.4$ % of the total $N_2$ fixation). In the *T. erythraeum* and in *C. watsonii* treatments, the proportion of DD$^{15}$N recovered in the dissolved pool was lower ($10.3 \pm 4.7$ % and $7.0 \pm 3.0$ % of the total $N_2$ fixation, respectively) but led to higher DD$^{15}$N transfer efficiencies ($11.7 \pm 4.4$ % and $5.8 \pm 2.7$ % of the total $N_2$ fixation, respectively). This suggests that the N compounds released by *Cyanothece* were less available for the surrounding plankton communities than the compounds released by *T. erythraeum* and *C. watsonii*.

On the opposite, the $^{15}NH_4^+$ enrichment appeared to be a relevant indicator of the DDN transfer efficiency: it was twice as high in the *T. erythraeum* treatment compared to the UCYN treatments (Table 2), leading to a DD$^{15}$N transfer efficiency twice as high in the *T. erythraeum* treatments (Table 1). This coupling between $^{15}NH_4^+$ enrichment and transfer efficiency suggests that $NH_4^+$ is the major form of DD$^{15}$N that is transferred to non-diazotrophic plankton, and that the DDN released under the form of DON is likely poorly available for the surrounding planktonic communities. This is in good agreement with the known higher bioavailability of $NH_4^+$ for phytoplankton compared to DON (e.g. Bradley et al., 2010; Collos and Berges, 2002). However, some DON compounds such as urea or amino acids can also be a significant source of N for planktonic communities, e.g. heterotrophic bacteria and mixotrophic plankton (Antia, 1991; Bronk, 2007). Unfortunately, the methodology used here can not asserts the importance of DON compared to $NH_4^+$ in the DDN transfer.

It should be noted that the increase of plankton biomass associated with non-diazotrophs in the present study cannot only be explained by the DD$^{15}$N provided by $N_2$ fixation within the time frame of the incubation (48 h). While the DD$^{15}$N trans-

Discussion Paper | Discussion Paper | Discussion Paper | Discussion Paper | Discussion Paper

**BGD**

doi:10.5194/bg-2015-607

**Transfer of diazotroph derived nitrogen**

H. Berthelot et al.

ferred to non-diazotrophic plankton biomass ranged between 1 and 10 nmol N L$^{-1}$ in the diazotrophs-amended treatments, the non-diazotrophic biomass increased from 90 to 160 nmol N L$^{-1}$ in the diazotrophs-amended treatments. This suggests that production was also stimulated by DDN fixed prior the incubations, that was thus not $^{15}$N labeled.

## 4.4 Plankton groups benefiting from the DDN

Bacterial biomass increased from 60 to 120 % after the addition of three diazotrophs; it was the plankton group which responded the most to the diazotrophs inoculations, whatever the treatment considered (Fig. 1 and Table S1). This is consistent with the high $^{15}$N-enrichment of bacteria cells in *T. erythraeum* and *C. watsonii* treatments compared to the control treatment (Fig. 6). In contrast, the high bacterial biomass increase observed in the *Cyanothece* treatment contrasts with the relatively low $^{15}$N-enrichment of bacterial individual cells measured in this treatment (Fig. 6). This suggests that, in the *T. erythraeum* and *C. watsonii* treatments, bacteria took advantage of the DD$^{15}$N released during the incubation, while in *Cyanothece* treatment, bacteria may have mainly relied on DDN fixed prior to the beginning of the incubation. This is consistent with the higher accumulation of DD$^{15}$N in the DON pool in the *Cyanothece* treatment compared to the two other treatments, indicating that the DON compounds released by *Cyanothece* are likely less bio-available for the planktonic community compared to those released by *T. erythraeum* and *C. watsonii*.

The presence of bacteria in *Trichodesmium* spp. colonies has been widely studied (Hewson et al., 2009; Hmelo et al., 2012; Nausch, 1996; Paerl et al., 1989; Rochelle-Newall et al., 2014; Sheridan et al., 2002). *Trichodesmium* spp. harbours high heterotrophic bacterial activity (Nausch, 1996; Tseng et al., 2005) and abundance is found to be at least two orders of magnitude higher in *Trichodesmium* spp. colonies than in surrounding waters (Sheridan et al., 2002). Associations between bacteria and UCYN are less documented. However, similarly to *Trichodesmium* spp., tight relationships may

**BGD**

doi:10.5194/bg-2015-607

**Transfer of diazotroph derived nitrogen**

H. Berthelot et al.

occur between UCYN and bacteria, as suggested by the significant increases of bacterial abundances in the UCYN treatments, but further investigations would be needed to understand the nature of their interactions.

Phytoplankton (diatoms, *Synechococcus* and small eukaryotes) was also stimulated by the diazotroph blooms, althougt to a lower extend compared to bacteria in all treatments. (Fig. 1). However, the increase in phytoplankton biomass in the *T. erythraeum* and *C. watsonii* treatments together with the significant $^{15}$N-enrichments in diatoms and *Synechococcus* (Fig. 6) argue that phytoplankton also took advantage of the DDN. This confirms the ability of diazotroph to promote non-diazotrophic primary producers as suggested by previous studies (e.g. Bonnet et al., 2015b; Devassy et al., 1979; Lee Chen et al., 2011; Lenes and Heil, 2010). The enhancement of large phytoplanktonic cells such as diatoms by DDN observed within the timespan of this study reveals the tight relashionship that may occurs between the new production fueled by diazotrophy and particle export in oligotrophic areas.

In the present study, the plankton community composition remained relatively stable in comparison to the Bonnet et al. (2015b) study, in which the authors observed systematic shifts from *Trichodesmium* spp. biomass towards diatom biomass. This difference is probably linked to the *Trichodesmium* cells status. In the Bonnet et al. (2015b) study, most of the analyzed *Trichodesmium* spp. cells were decaying, leading to the release of micromolar concentrations of $NH_4^+$ accumulating in the dissolved pool, whereas the main form of DDN released in the present study was DON. This rapid increase in $NH_4^+$ bio-availability was benefiting to diatoms, which are known to be highly competitive under high nutrient concentrations (Chavez and Smith, 1995; Kudela and Dugdale, 2000; Smetacek, 1998; Wilkerson et al., 2000). This synchronized destruction of the colonies has been shown to be possible within a few hours, mediated by programmed cell death (Berman-Frank et al., 2004). Furthermore, the dense bloom forming behavior and maintenance of *Trichodesmium* spp. at the surface due to their positive buoyancy (Romans et al., 1994) may be an additional feature that helps the constitution of locally rich N layers promoting the diatom development. In the present study, the *T. erythraeum*

**BGD**

doi:10.5194/bg-2015-607

**Transfer of diazotroph derived nitrogen**

H. Berthelot et al.

colonies added to the incubation bottles were in exponential growing phase and the relative stability of their abundance throughout the experiment indicates no substantial cell breakages. The amount of DDN transfer in our study was thus mainly mediated by the release of $NH_4^+$ and DON during active $N_2$ fixation rather than the N release due to cell breakage and was thus much more limited than in the Bonnet et al. (2015b) study, leading to attenuated changes in the planktonic community composition.

## 5 Conclusions and ecological implications

This study reveals the various short term fates of DDN in the ocean and highlight the complex interactions between diazotrophs and their environment. First, it shows that the DDN released by diazotrophs in the dissolved pool as $NH_4^+$ is quickly transferred to non-diazotrophic plankton while the DDN released as DON is mostly accumulated in the dissolved pool. Second, the DDN transfer efficiency towards the non-diazotrophic plankton depends on the diazotrophs involved in $N_2$ fixation: it is twice as much for *T. erythraeum* compared to the DDN transfer associated with UCYN strains. This implies that *T. erythraeum* would be more efficient at promoting non-diazotrophic marine productivity in N-depleted areas than UCYN are. Finally, the results presented here suggest that diazotrophic activity first promotes heterotrophic plankton but also autotrophic plankton, albeit to a lower extent. Taken together, theses results show that the fates of DDN are diverse and would need further investigation, in particular in the vast open-ocean regions where primary productivity extensively depends on diazotrophy.

*Author contributions.* S. Bonnet and H. Berthelot designed the experiments and S. Bonnet and H. Berthelot carried them out. All authors analyzed the samples. H. Berthelot prepared the manuscript, which was amended by S. Bonnet.

*Acknowledgements.* Funding for this research was provided by the Agence Nationale de la Recherche (ANR starting grant VAHINE ANR-13-JS06-0002). We thank François Robert, Smail Mostefaoui and Rémi Duhamel from the French National Ion MicroProbe Facility hosted by the

**BGD**

doi:10.5194/bg-2015-607

**Transfer of diazotroph derived nitrogen**

H. Berthelot et al.

Museum National d'Histoire Naturelle (Paris) for providing nanoSIMS facilities and constant advice. We are grateful to Gerald Gregori from the Regional Flow Cytometry Platform for Microbiology (PRECYM) of the Mediterranean Institute of Oceanography (MIO) for the flow cytometry analyses support.

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

**Table 1.** Synthesis of the distribution of the recently fixed $N_2$ ($DD^{15}N$) in each of the planktonic groups analysed after 48 h of incubation (nmol L$^{-1}$ 48 h$^{-1}$) and their respective proportion relative to the total fixed $N_2$ (%). n.a.: not analysed, n.c.: not calculated. Standard deviations are in parenthesis.

| Planktonic group | $DD^{15}N$ (nmol L$^{-1}$ 48 h$^{-1}$) | % of total $N_2$ fixed |
|---|---|---|
| *Trichodesmium* treatment | | |
| *Trichodesmium* | 63.05 (21.54) | 78.0 (26.7) |
| Dissolved pool | 8.30 (3.83) | 10.3 (4.7) |
| Sum of non-diazotrophs | 9.45 (3.54) | 11.7 (4.4) |
| Bacteria | 5.79 (3.41) | 7.2 (4.2) |
| Diatom | 1.99 (0.36) | 2.5 (0.5) |
| *Synechococcus* | 1.67 (0.88) | 2.1 (1.1) |
| Small eukaryotes | n.a | n.c |
| *C. watsonii* treatment | | |
| *C. watsonii* | 60.92 (10.06) | 87.2 (14.4) |
| Dissolved pool | 4.90 (14.10) | 7.0 (3.0) |
| Sum of non-diazotrophs | 4.08 (1.87) | 5.8 (2.7) |
| Bacteria | 2.36 (1.70) | 3.4 (2.4) |
| Diatom | 0.06 (0.05) | 0.1 (0.1) |
| *Synechococcus* | 1.63 (0.78) | 2.3 (1.1) |
| Small eukaryotes | 0.03 (0.00) | n.c. |
| *Cyanothece* treatment | | |
| *Cyanothece* | 19.41 (5.28) | 78.6 (21.4) |
| Dissolved pool | 4.10 (9.85) | 16.6 (5.0) |
| Sum of non-diazotrophs | 1.19 (0.59) | 4.9 (2.4) |
| Bacteria | −0.05 (0.42) | n.c. |
| Diatom | 0.93 (0.28) | 3.8 (1.1) |
| *Synechococcus* | 0.29 (0.30) | 1.2 (1.2) |
| Small eukaryotes | 0.02 (0.00) | n.c. |

Discussion Paper | Discussion Paper | Discussion Paper | Discussion Paper

**BGD**

doi:10.5194/bg-2015-607

**Transfer of diazotroph derived nitrogen**

H. Berthelot et al.

**BGD**

doi:10.5194/bg-2015-607

**Transfer of diazotroph derived nitrogen**

H. Berthelot et al.

**Table 2.** $^{15}$N-enrichment (atom %) of diazotrophic cells, PON, $NH_4^+$ and DON pools. In parenthesis are shown the standard deviations on triplicate incubations. n/a: not applicable, n.d.: not detected.

| | Control | *T. erythraeum* treatment | *C. watsonii* treatment | *Cyanothece* treatment |
|---|---|---|---|---|
| Diazotrophic cells | n/a | 1.15 (0.23) | 1.94 (0.24) | 2.50 (0.30) |
| PON | 0.46 (0.01) | 0.81 (0.13) | 0.72 (0.01) | 0.58 (0.01) |
| $NH_4^+$ | n.d. | 2.31 (0.81) | 1.20 (0.15) | 1.44 (0.44) |
| DON | 0.37 (< 0.00) | 0.37 (< 0.00) | 0.37 (< 0.00) | 0.38 (< 0.00) |

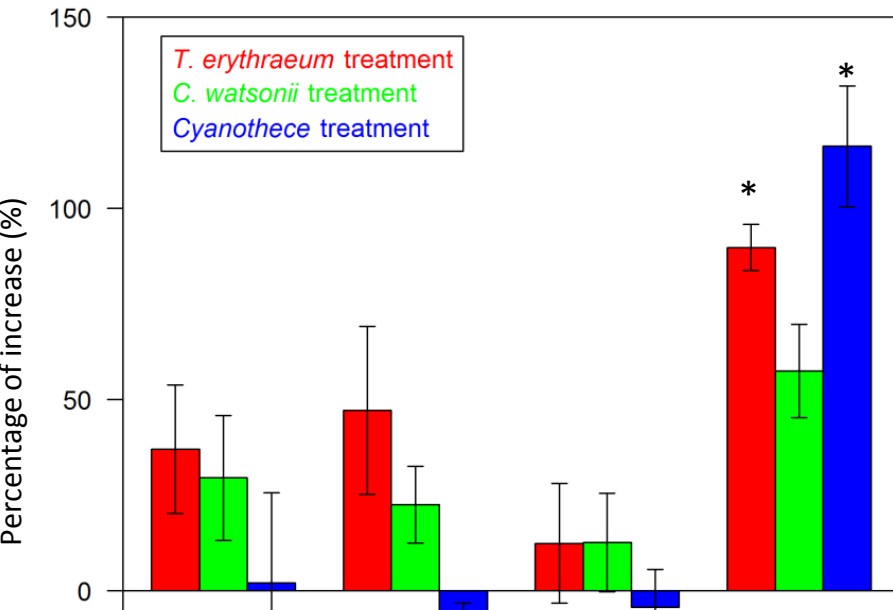

**Figure 1.** Relative increase of biomass associated with non-diazotrophic plankton groups considered in this study in the three diazotrophs-amended treatments relative to the control (%) after 48 h of incubation. Errors bar represent the standard deviations on triplicate incubations of both diazotrophs-amended treatments and control treatment. * depict significant increase in biomass (unpaired Tukey HSD test, at 95 % levels of confidence).

**BGD**

doi:10.5194/bg-2015-607

**Transfer of diazotroph derived nitrogen**

H. Berthelot et al.

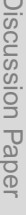

**BGD**

doi:10.5194/bg-2015-607

**Transfer of diazotroph derived nitrogen**

H. Berthelot et al.

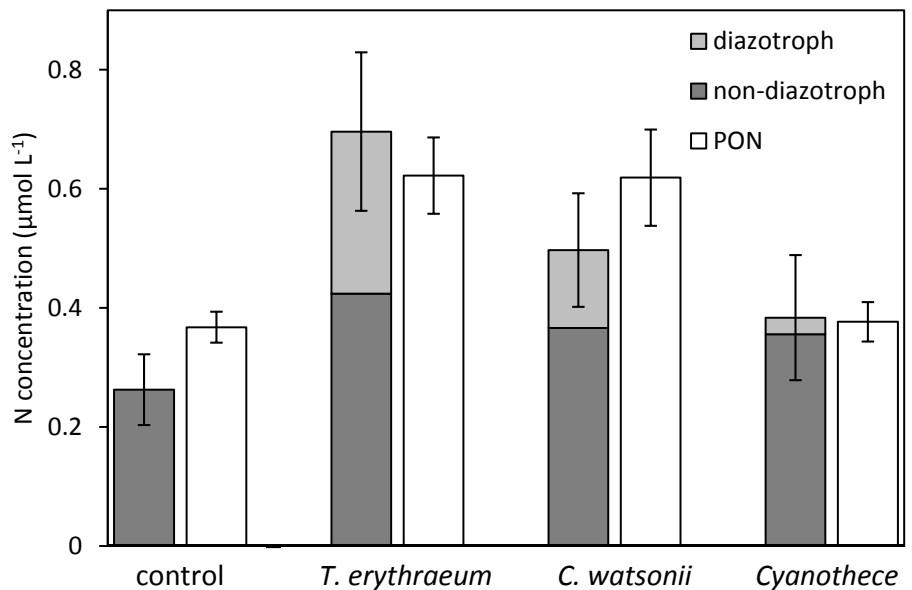

**Figure 2.** PON concentrations measured by mass spectrometry (EA-IRMS) and biomass associated with each plankton group in each treatment after 48 h of incubation ($\mu mol\,L^{-1}$). Errors bar represent the standard deviation on triplicate incubations.

Discussion Paper | Discussion Paper | Discussion Paper | Discussion Paper | Discussion Paper |

**BGD**

doi:10.5194/bg-2015-607

**Transfer of diazotroph derived nitrogen**

H. Berthelot et al.

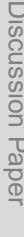

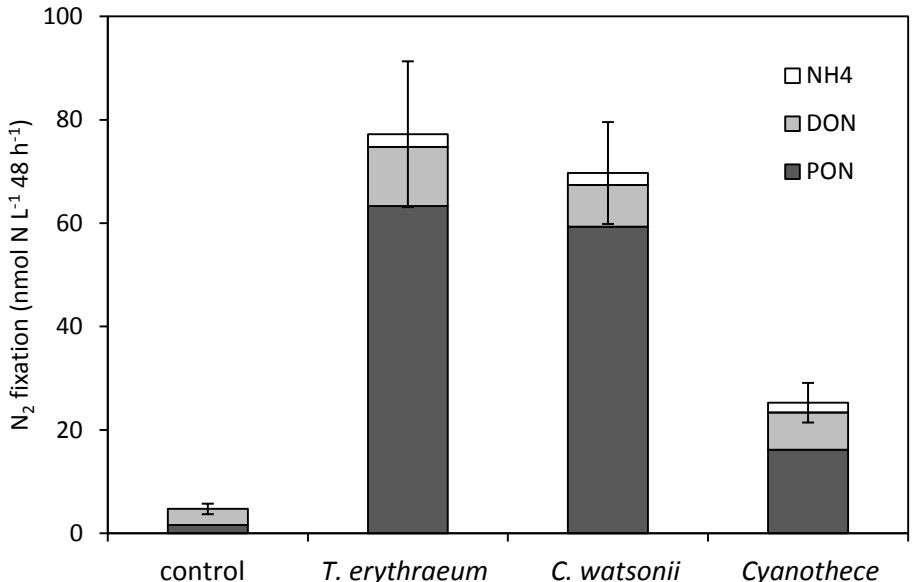

**Figure 3.** $N_2$ fixation rates (dark grey, $\mathrm{nmol\,L^{-1}\,48\,h^{-1}}$), DDN release ($\mathrm{nmol\,L^{-1}\,48\,h^{-1}}$) as DON (light grey) and $NH_4^+$ (white) in each treatment. Error bars represent the standard deviation of triplicate incubations and the propagated analytical error.

Discussion Paper | Discussion Paper | Discussion Paper | Discussion Paper |

**BGD**

doi:10.5194/bg-2015-607

**Transfer of diazotroph derived nitrogen**

H. Berthelot et al.

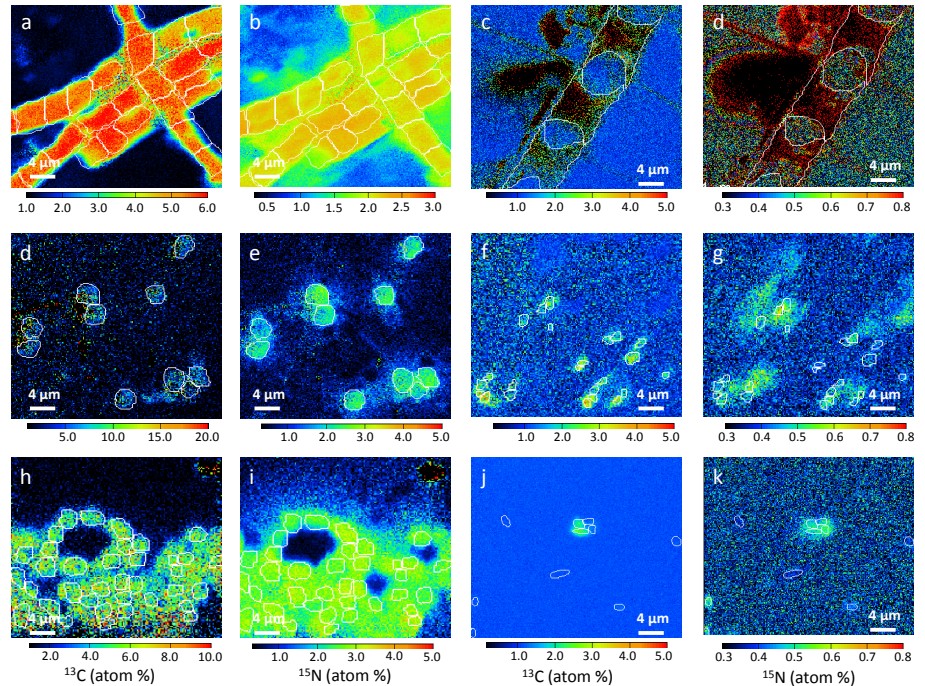

**Figure 4.** NanoSIMS images showing the isotopic enrichment found in cells after 48 h of incubation. $^{13}$C- **(a, d, h, c, f, j)** and $^{15}$N- **(b, e, i, d, g, k)** enrichments (atom %) are shown for *T. erythraeum* **(a, b)**, *C. watsonii* **(d, e)**, *Cyanothece* **(h, i)**, *Chaetoceros* sp. **(c, d)**, *Synechococcus* **(f, g)** and bacteria cells **(j, k)**. The white outlines define the regions of interest (ROIs), which were used to estimate the cells $^{13}$C- and $^{15}$N-enrichments.

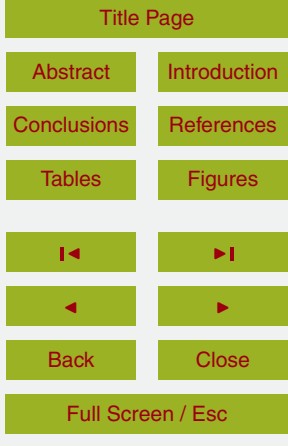

**Figure 5.** $^{15}$N-enrichment (atom %) measured in *T. erythraeum* (red), *C. watsonii* (green) and *Cyanothece* (blue) cells relative to the $^{13}$C enrichment. Box plots of $^{13}$C- and $^{15}$N-enrichments are shown, following the same color code, on horizontal and vertical axes, respectively.

**BGD**

doi:10.5194/bg-2015-607

**Transfer of diazotroph derived nitrogen**

H. Berthelot et al.

Discussion Paper | Discussion Paper | Discussion Paper | Discussion Paper

**BGD**

doi:10.5194/bg-2015-607

**Transfer of diazotroph derived nitrogen**

H. Berthelot et al.

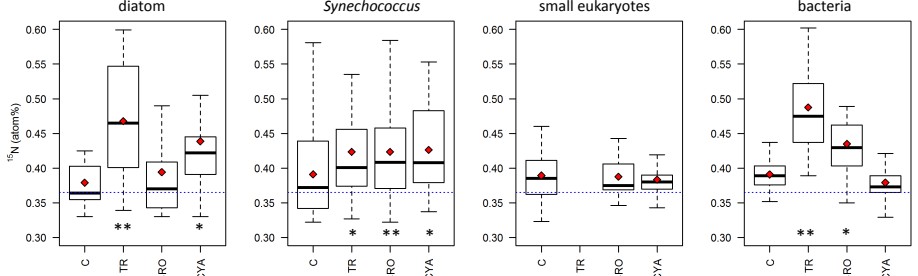

**Figure 6.** Box-plot of $^{15}$N-enrichment measured in diatoms, *Synechococcus*, small eukaryotes and bacteria in the control (C), *T. erythraeum* (TR), *C. watsonii* (CRO) and *Cyanothece* (CYA) treatments. Red dots indicate the average values. Blue dotted lines depict the natural $^{15}$N abundance. * and ** depict significant enrichments in diazotrophs treatments compared to the control treatment (unpaired Mann–Whitney–Wilcoxon test) at 95 and 99 % levels of confidences, respectively.