# Peer review of "Transfer of diazotroph derived nitrogen towards non-diazotrophic planktonic communities: a comparative study between *Trichodesmium erythraeum*, *Crocosphaera watsonii* and *Cyanothece* sp."

_Biogeosciences, 2015_

## Referee Comment (RC1) · Anonymous Referee #1 · 23 Feb 2016

This study examined the fate of nitrogen fixed by three major cyanobacterial diazotrophs (Trichodesmium erythraeum, Crocosphaera watsonii, Cyanothece sp.) in the tropical and subtropical oligotrophic regions by microcosm experiments. Nitrogen transfer from individual diazotroph groups to the host organism or non-diazotrophs in the ambient water has been examined (Lee Chen et al., 2011; Foster et al., 2011; Krupke et al., 2015), however, this kind of comparative study is lacking. Therefore, if the finding in the present study is true, it will be of interest to the journal's readership.

The focus of the paper is well phrased, and the manuscript is well written. However, the authors should make clear the following points before the publication.

The authors evaluated fixed N and N transfer from 15N signal at the cell surface. This analysis would not be examined fixed N inside the cell. The authors have to justify their "quantitative" analysis especially in this kind of comparative study. Otherwise, their conclusion is not very convincing.

Ammonium concentration is generally depleted in the oligotrophic ocean because it is rapidly consumed by microorganism as soon as it is released (Brzezinski, 1988). The authors should show the detection limit of ammonium concentration in their analysis of 15N-ammonium. Further, they should show ammonium and DON concentration in each experiment in the table.

Why was the fixed N transferred mainly towards pico-planktonic communities? In general, when light and nutrient are sufficiently supplied, diatoms are capable of growing more rapidly than cyanobacteria (Miller and Wheeler, 2012).

Specific P21 L2-4 How did the authors calculate the cell-specific N2 fixation by using a nanoSIMS? Please write more detail.

P26 L16-21 Bonnet et al. (2015b) is unpublished paper, and thus I cannot evaluate this discussion.

Reference Brezezinski, M.A. (1988) Vertical distribution of ammonium in stratified oligotrophic waters., Limnol. Oceanogr. 33(5), 1176-1182. Krupke, A., W. Mohr, J. LaRoche, B.M. Fuchs, R.I. Amann, M.M.M. Kuypers (2015) The effect of nutrients on carbon and nitrogen fixation by the UCYN-A-haptophyte symbiosis., ISME J. 9, 1635-1647. Miller, C.B., P.A. Wheeler (2012) Chapter 3 Habitat determinants of primary production in the sea. In: Biological Oceanography, Second Edition, John Wiley & Sons, Ltd.

---

## Referee Comment (RC2) · Anonymous Referee #2 · 10 Mar 2016

**General comments**

This study used $^{15}$N and $^{13}$C labeling coupled with cell sorting by flow cytometry and nanoSIMS analyses to look at the transfer of $^{15}$N-labeled diazotroph-derived N (DDN) to the dissolved pool and non-diazotrophs. The authors used a microcosm experiment to compare the transfer efficiency of $^{15}$N-labeled DDN for different diazotroph groups: *Trichodesmium erythraeum*, *Crocosphaera watsonii* and *Cyanothece* sp.

I found these results extremely interesting and timely. They elucidate the fate of fixed N from $N_2$-fixation as well as the specific role of different diazotrophic communities in the ocean. The paper is clear and well written. I recommend publication after minor revisions (see below).

**Specific comments:**

**Abstract:**

Page 3, line 28: "…heterotrophic bacteria followed phytoplankton… ". Do they mean: "heterotrophic bacteria followed by phytoplankton"?

**Introduction**

Page 5, line 8: Bourbonnais et al. (2009) also observed low $\delta^{15}$N-PON in sediment traps (most likely from $N_2$ fixation in surface waters) in the subtropical northeast Atlantic and should also be cited here.

Page 5, line 16: I do not believe that UCYN-C (unicellular cyanobacterial Group C) is defined previously.

**Materials and methods:**

Page 9, line 12: I think a few lines should be added regarding the potential contamination of $^{15}N_2$ gas by $^{15}$N-labeled $NO_3^-$, $NO_2^-$ and $NH_4^+$, that could lead to overestimation of $N_2$ fixation rates, as reported by Dabundo et al. (2014). Although two batch syntheses of the Cambridge Isotopes gas were determined to contain only trace concentrations of $^{15}$N $NH_4^+$, $NO_2^-$ and $NO_3^-$, I am curious to know if the authors verified the purity of the $^{15}N_2$ gas used before their experiment.

Page 9, lines 18-19: How and how long was the bag ($^{15}N_2$ bubble) shaken?

Page 12, lines 19-21: $^{15}$N depleted $NO_3^-$ (likely from $N_2$ fixation) was observed in the subtropical north Atlantic Ocean (see Knapp et al., 2008; Bourbonnais et al., 2009). Since nitrification can occur in the euphotic zone (Yool et al., 2007), it is thus possible for part of the labeled $^{15}$N pool to be transferred to the NOx pool (particularly $NO_3^-$), which could then be rapidly assimilated. I agree that the contribution from $NH_4^+$ should be more

significant, and that this mechanism would be more important at lower irradiance deeper in the water column, but I think this point should, at least, be discussed.

**Results:**

Page 18, lines 21: why would the DDN be higher (at least double) in the control treatment?

**Discussion and conclusions**
Page 20, lines 18-26: This whole paragraph is a repetition of the introduction. I would remove.

Page 21, line 10: Can these rates be compared with the one in Garcia et al. (2007)? $N_2$ fixation rates using methods prior to the one developed by Mohr et al. (2010) tend to be underestimation, whereas rates calculated with contaminated gas stocks (Dabundo et al., 2014) tend to be overestimation.

Page 23, lines 15-23: Can the authors explain what may cause the significant differences in $DD^{15}N$ transfer in the UCYN treatments observed in their study compared to Bonnet et al. (2015a)?

Page 24, lines 19-22: This is also in agreement with the observation of a recalcitrant DON pool by Knapp et al., 2005 and Bourbonnais et al. (2009) in the subtropical Atlantic, on the basis of the concentration of DON its $\delta^{15}N$ in surface water.

Page 25, line 4: What were the [DON] and [$NH_4^+$] concentrations prior to the incubations?

Page 26, starting line: Bonnet et al. (2015b) is currently in review, making it difficult to evaluate this part of the discussion. Please update.

**Tables and Figures**

Table 2: DON, $NH_4^+$, $NO_2^-$ and $NO_3^-$ concentrations should also be included in the table for the different treatments.

Figure 5. Please show significant linear regressions, with $r^2$ and p-value.

Figure 6: This figure is too small. The font for x- and y-axis should be increased as well as the size of the overall figure.

**Technical corrections**

Page 5, line 4: replace the second "which" by "and".

Page 11, line 16: replace "chlrorphyll" by "chlorophyll"

Page 17, line 10: replace "to" by "the"

**References:**

Bourbonnais, A., M. F. Lehmann, J. J. Waniek, and D. E. Schulz-Bull (2009), Nitrate isotope anomalies reflect $N_2$ fixation in the Azores Front region (subtropical NE Atlantic), *J. Geophys. Res., 114*, C03003, doi:10.1029/2007JC004617.

Dabundo, R., M. F. Lehmann, L. Treibergs, C. R. Tobias, M. A. Altabet, P. H. Moisander, and J. Granger (2014), The contamination of commercial $^{15}N_2$ gas stocks with $^{15}N$-labeled nitrate and ammonium and consequences for nitrogen fixation measuremetns, PLOS one, 9, 10, e110335.

Knapp, A.N., P.J. DiFiore, C. Deutsch, D.M. Sigman, and F. Lipschultz (2008), Nitrate isotopic composition between Bermuda and Puerto Rico Implications for $N_2$ fixation in the Atlantic Ocean, *Global Biogeochem. Cy., 22*, GB3014.

Yool, A., A. P. Martin, C. Fernández, and D. R. Clark (2007), The significance of nitrification for oceanic new production, *Nature, 447*, 999-1002.

---

## Author Comment (AC1) · 17 May 2016

**Authors's response**

We are grateful to the referees for their useful comments which have significantly increased the quality of the paper. The comments are addressed below (referee comments are in bold and authors answers are in regular font).

**Anonymous Referee #1**

***The authors evaluated fixed N and N transfer from 15N signal at the cell surface. This analysis would not be examined fixed N inside the cell. The authors have to justify their "quantitative" analysis especially in this kind of comparative study.   Otherwise, their conclusion is not very convincing.***

Prior to analysis, cell surface is pre-sputtered with a powerful beam that removes the surface layers of the cells.  Prior to each analysis session, the beam energy is measured and the pre-sputtering time is adapted to the analyzed cell characteristics (basically, a longer pre-sputtering is performed for larger cells and silica diatom frustules). Thus, the $^{15}N/^{14}N$ ratio is determined from inner cellular N. We modified the text as follows on page 14, line 14:  "Samples were pre-sputtered prior to analyses with a current of ~10 pA for at least 2 min to achieve sputtering equilibrium and insure the analysis to be performed inside the cells by removing cell surface."

***Ammonium concentration is generally depleted in the oligotrophic ocean because it is rapidly consumed by microorganism as soon as it is released (Brzezinski, 1988).  The authors should show the detection limit of ammonium concentration in their analysis of 15N-ammonium. Further, they should show ammonium and DON concentration in each experiment in the table.***

DON and ammonium concentrations are now included in Table 2.

Regarding the $^{15}NH_4$, it is not possible to give a detection limit as the $^{15}N$ enrichment both depends on the $^{15}N$ enrichment of the $NH_4$ pool and the $NH_4$ concentration. Thus, two situations may lead to undetect the $^{15}NH_4$ enrichment. First, a concentration in $NH_4$ below the quantification limit (basically ~9 nmol $L^{-1}$) despite measurable $^{15}N$ enrichment on mass spectrometer. Second, a measurable $NH_4$ concentration, but a $^{15}N$ enrichment below three times the standard deviation of the blank values (used as quantification limit of the $^{15}N$ enrichment). This latter case explains the absence of $^{15}NH_4$ enrichment in the control treatment.

***Why was the fixed N transferred mainly towards pico-planktonic communities? In general, when light and nutrient are sufficiently supplied, diatoms are capable of growing more rapidly than cyanobacteria (Miller and Wheeler, 2012).***

This question is discussed in section 4.4, from page 26, line 15 to page 27, line 6. The Miller and Wheeler, 2012 reference has been added to the text on page 26, line 24.

*Specific*

*P21 L2-4 How did the authors calculate the cell-specific N2 fixation by using a nanoSIMS? Please write more detail.*

Cell-specific $N_2$ fixation rates using nanoSIMS were calculated according the equation (1) (page 16). We acknowledge that this was not clearly explicated and we modified the following section on page 16, line 1 : "The DD$^{15}$N_cell-specific $N_2$ fixation_and transfer (that depict the amount of $^{15}N_2$ transferred from diazotrophs towards the non-diazotrophic plankton) were expressed in nmol L$^{-1}$ 48 h$^{-1}$ and calculated for each plankton group analysed as follows:…."

*P26 L16-21 Bonnet et al. (2015b) is unpublished paper, and thus I cannot evaluate this discussion.*

The Bonnet et al. (2015b) is available on request (contact: sophie.bonnet@uni-amu.fr). The article is currently *in press* and should be available soon on publisher's web page.

**Anonymous Referee #2**

*Specific comments:*

*Abstract:*

*Page 3, line 28: "…heterotrophic bacteria followed phytoplankton… ". Do they mean: "heterotrophic bacteria followed by phytoplankton"?*

Indeed, we meant "heterotrophic bacteria followed by phytoplankton". The change has been applied.

*Introduction*

*Page 5, line 8: Bourbonnais et al. (2009) also observed low δ15N-PON in sediment traps (most likely from N2 fixation in surface waters) in the subtropical northeast Atlantic and should also be cited here.*

The reference has been added.

*Page 5, line 16: I do not believe that UCYN-C (unicellular cyanobacterial Group C) is defined previously.*

The sentence has been re written and UCYN-C are now defined as follows: "…..mainly related to group C….."

*Materials and methods:*

*Page 9, line 12: I think a few lines should be added regarding the potential contamination of 15N2gas by 15N-labeled NO3-, NO2-and NH4+, that could lead to overestimation of N2 fixation rates, as reported by Dabundo et al. (2014). Although two batch syntheses of the Cambridge Isotopes gas were determined to contain only trace concentrations of 15N NH4+, NO2-and NO3,*

*I am curious to know if the authors verified the purity of the 15N2 gas used before their experiment.*

The purity of the batch has been verified by Dabundo's group. It appeared the $^{15}N_2$ bottles contained very low levels of contamination, leading to rates ranging from undetectable to 0.04 nmol $L^{-1}$ 48 $h^{-1}$. This would represent less than 0.05 % of the calculated DDN, and was thus neglected. Nevertheless, in order to clarify, we added in the method section the following sentence: "The potential contamination by $^{15}NOx$ and $^{15}NH_3$ of the $^{15}N_2$ bottles, recently highlighted by Dabundo et al. (2014), was tested on one of our $^{15}N_2$ Cambridge Isotope batch. According to the model described in Dabundo et al. (2014), it appeared that the low level of contamination measured (1.4 10-8 mol of $^{15}NO_3$ $mol^{-1}$ of $^{15}N_2$ and 1.1 10-8 mol $NH_4^+$ $mol^{-1}$ of $^{15}N_2$) would only contribute to ~0.05 % of the $DD^{15}N$ measured in our study and was thus neglected."

*Page 9, lines 18-19: How and how long was the bag (15N2 bubble) shaken?*

The bag was shaken until the complete dissolution of the bubble; basically it took from 5 to 10 minutes. The duration is now mentioned in the text on page 9, line 18: "The $^{15}N_2$ bubble was vigorously shaken for 5 to 10 minutes until its complete dissolution."

*Page 12, lines 19-21: 15N depleted NO3- (likely from N2 fixation) was observed in the subtropical north Atlantic Ocean (see Knapp et al., 2008; Bourbonnais et al., 2009). Since nitrification can occur in the euphotic zone (Yool et al., 2007), it is thus possible for part of the labeled 15N pool to be transferred to the NOx pool (particularly NO3-), which could then be rapidly assimilated. I agree that the contribution from NH4+ should be more significant, and that this mechanism would be more important at lower irradiance deeper in the water column, but I think this point should, at least, be discussed.*

We agree that nitrification may play a role, but is assumed to be negligible (Raimbault and Garcia, 2008). The possibility of surface nitrification is now mentioned as follows on page 12, ligne 19: "Furthermore, they were unlikely to be released by diazotrophs, thus unlikely $^{15}N$-enriched. Nitrification, that converts $NH_4^+$ to $NO_3^-$ at rates rising 5-10 nmol $L^{-1}$ $d^{-1}$ in N depleted surface waters (e.g. Yool et al. 2007) may have contributed to the underestimation of the transfer of $DD^{15}N$ in the $NH_4^+$ pool and to an overestimation of the $DD^{15}N$ in the DON pool. Nevertheless, in surface water, nitrification fluxes are found to be several orders of magnitude lower than $NH_4^+$ regeneration (Raimbault and Garcia, 2008) and were thus neglected in the interpretation of the results"

*Results:*

*Page 18, lines 21: why would the DDN be higher (at least double) in the control treatment?*

The proportion of DDN release in the dissolved pool compared to the total $N_2$ fixation is indeed higher in the control treatment. However, the absolutes rates are lower in the control treatment compared to others treatments. In other words, the addition of diazotrophs in the natural planktonic community increased the amount of DDN released in the dissolved pool but to a lower proportion to the release performed by the diazotrophs initially present in the planktonic community. It is possible that the lower proportion of release that displays the cultured

diazotrophs is due to their life cycle status. This is discussed in section 4.2 (from page 22, line 12 to page 23 line 4).

***Discussion and conclusions***

***Page 20, lines 18-26: This whole paragraph is a repetition of the introduction. I would remove.***

Despite the repetition of the introduction, we believe that this short paragraph contextualize the discussion and facilitate the reading. Nevertheless, we reduce the length of this section as follows: "The fate of the DDN in the marine food web has been poorly studied, mainly due to technical limitations. Using $^{15}$N and $^{13}$C labeling coupled with cell sorting by flow cytometry and nanoSIMS analyses at the single cell level, we were able to trace the transfer of DD$^{15}$N from the diazotrophs to the dissolved pool and to the non-diazotrophic plankton, and compare the DD15N transfer efficiency as a function of the diazotroph groups dominating the community."

***Page 21, line 10: Can these rates be compared with the one in Garcia et al. (2007)? N2 fixation rates using methods prior to the one developed by Mohr et al. (2010) tend to be underestimation, whereas rates calculated with contaminated gas stocks (Dabundo et al., 2014) tend to be overestimation.***

It is true that recent methodological improvements/insights tended to question the accuracy of former $^{15}$N$_2$ based N$_2$ fixation rates. However, we are confident that the orders of magnitude of N$_2$ fixation rates reported in former studies confirm the presence of relatively high N$_2$ fixation rates (range 10-100 nmol L-1 d-1) in this region (Berthelot et al., *Biogeosciences* (vol 12 (13)) 2015, Bonnet et al., *Global Biogeochemical Cycles* (vol 29 (11)), 2015).

***Page 23, lines 15-23: Can the authors explain what may cause the significant differences in DD15N transfer in the UCYN treatments observed in their study compared to Bonnet et al. (2015a)?***

The following sentence has been added in order to explain the discrepancy between Bonnet et al. (2015a) and our study on page 23, line 23: "The DD$^{15}$N transfer reported in the latter study is ~three times higher than in the present study. This discrepancy may result from the physiological differences between the UCYN-C ecotypes involved in both studied, and/or from the DDN release from diazotrophic cells that is potentially higher is natural communities compared to cultured cells as discussed above."

***Page 24, lines 19-22: This is also in agreement with the observation of a recalcitrant DON pool by Knapp et al., 2005 and Bourbonnais et al. (2009) in the subtropical Atlantic, on the basis of the concentration of DON its δ15Nin surface water.***

Knapp et al., 2005 and Bourbonnais et al. (2009) references have been added in the revised version of the manuscript on page 24, line 20: "……and that the DDN released under the form of DON is likely poorly available for the surrounding planktonic communities (Knapp et al., 2005, Bourbonnais et al., 2009)".

***Page 25, line 4: What were the [DON] and [NH4+] concentrations prior to the incubations?***

The concentrations are now shown in Table 2.

***Page 26, starting line: Bonnet et al. (2015b) is currently in review, making it difficult to evaluate this part of the discussion. Please update.***

The Bonnet et al. (2015b) is available on request mailing to sophie.bonnet@uni-amu.fr. The article is currently *in press* and should be available soon on publisher's web page.

***Tables and Figures***

***Table 2: DON, NH4+, NO2-and NO3- concentrations should also be included in the table for the different treatments.***

DON and $NH_4$ concentrations are now shown in Table 2. $NO_2$ and $NO_3$ were only measured at the time of sampling and were found to be <0.2 μmol $L^{-1}$. The latter concentrations are mentioned in the method section (page 12, line 19).

***Figure 5. Please show significant linear regressions, with r2 and p-value.***

The regressions are now shown in the new version of Fig. 5. In addition, the caption has been modified as follows: "$^{15}N$-enrichment (atom%) measured in *T. erythraeum* (red), *C. watsonii* (green) and *Cyanothece* (blue) cells relative to the 13C enrichment. The colored line are the linear regressions for *T. erythraeum* (red), *C. watsonii* (green) shown with their respective r-squared and p-values. Regression is not significant for *Cyanothece* and thus not shown on the plot. Box plots of $^{13}C$- and $^{15}N$-enrichments are shown, following the same color code, on horizontal and vertical axes, respectively."

***Figure 6: This figure is too small. The font for x- and y-axis should be increased as well as the size of the overall figure.***

The size of the figure and the font has been increased (see modified figures at the end of the document).

***Technical corrections***

***Page 5, line 4: replace the second "which"by "and".***

***Page 11, line 16: replace "chlrorphyll" by "chlorophyll"***

These changes have been done

Additional changes:

In addition, in the new version of the manuscript, we correct few mistakes that were present in the first version:

-page 13, line 1:  detection limit = 3 nmol $L^{-1}$

-Table 2: The [15]N-enrichment (atom%) of the PON were pool are 0.40, 0.69, 0.67 and 0.50 for control, *T. erythraeum*, *C. watsonii* and *Cyanothece* treatments instead of 0.46, 0.81, 0.72, 0.58.

Table 2, Figure 5 and Figure 6 has been modified as follows:

Table 2. [15]N-enrichment (atom%) of diazotrophic cells, PON, $NH_4^+$ and DON pools and concentrations ($\mu$mol $L^{-1}$) of $NH_4^+$ and DON. In parenthesis are shown the standard deviations on triplicate incubations. n.a.: not applicable, n.d.: not detected.

| | Control | *T. erythraeum* treatment | *C. watsonii* treatment | *Cyanothece* treatment |
|---|---|---|---|---|
| | | [15]N-enrichment | | |
| Diazotrophic cells | n.a. | 1.15 (0.23) | 1.94 (0.24) | 2.50 (0.30) |
| PON | 0.40 (0.01) | 0.69 (0.13) | 0.67 (0.01) | 0.50 (0.01) |
| $NH_4^+$ | n.d. | 2.31 (0.81) | 1.20 (0.15) | 1.44 (0.44) |
| DON | 0.37 (<0.00) | 0.37 (<0.00) | 0.37 (<0.00) | 0.38 (<0.00) |
| | | Concentrations | | |
| $NH_4^+$ | 0.010 (0.002) | 0.010 (0.003) | 0.011 (0.004) | 0.009 (0.003) |
| DON | 4.05 (0.57) | 3.99 (0.17) | 4.54 (0.80) | 4.18 (0.45) |

[Figure]

Figure 5. ¹⁵N-enrichment (atom%) measured in *T. erythraeum* (red), *C. watsonii* (green) and *Cyanothece* (blue) cells relative to the ¹³C enrichment. The colored line are the linear regressions for *T. erythraeum* (red), *C. watsonii* (green) shown with their respective r-squared and p-values. Regression is not significant for *Cyanothece* and thus not shown on the plot. Box plots of ¹³C- and ¹⁵N-enrichments are shown, following the same color code, on horizontal and vertical axes, respectively.

[Figure]

Figure 6.